# A Shot-Efficient Differential Equation Integrator using Quantum Neural Networks

## Abstract

Physics-informed regularisation on quantum neural networks provides a promising means for solving differential equations on near-term quantum computers. However, most demonstrations of this technique assume idealised simulated quantum circuits where the respective expectations are available. In real quantum hardware, such ideal expectations are not accessible and must be averaged over many shots, introducing additional computations, the cost of which has not been considered in the majority of the preceding studies. The requirements of higher-order derivatives for physics-informed regularisers are especially high in terms of circuit repetitions (shots) compared to lower-order derivatives required for supervised learning. We demonstrate how to construct a global formulation of physics-informed losses especially amenable to solve ordinary differential equations on near-term quantum computers in a shot-efficient manner. The resulting approach can reduce the order of derivatives required to calculate a loss compared to Physics-informed Neural Networks (PINNs). In the case of initial value problems in ordinary differential equations (ODEs) and some partial differential equations (PDEs), our method removes completely the need for higher-order automatic differentiation, thus providing an $\mathcal{O}(N)$ improvement in shot-efficiency, where $N$ is the number of data-encodings of the quantum neural network. Our formulation naturally incorporates boundary conditions and physics-informed losses into a single optimisation term. Numerical experiments demonstrate favourable empirical performance, in terms of both shot-efficiency and error, on (simulated) quantum circuits compared to existing quantum methodologies. We demonstrate that the relative performance of quantum neural network algorithms in the infinite shot limit does not necessarily correspond to relative performance in the finite shot limit. We hope this works provides insights on how to efficiently design schemes that will reduce the shot requirements and will become the basis for further developing efficient quantum algorithms for the solution of differential equations.

## 1 Introduction

Differential equations form the basis of modelling a great number of systems, such as electromagnetic waves, evolution of heat distributions, population dynamics, optimisation of industrial processes, and time-evolution of probability distributions. Consequently, there has been much recent interest on biasing neural networks towards differential equation solutions.

A common means of representing differential equations in neural networks is using Physics-informed neural networks (PINNs) Lagaris et al. (1998); Raissi et al. (2019), which use (potentially higher-order) derivatives of neural networks outputs with respect to their inputs to construct appropriate regularisation terms towards a prescribed differential equation.

In addition to their application to neural networks in conventional silicon-based hardware, physics-informed loss functions represent a promising means to solve differential equations using quantum neural networks (QNNs) on near-term quantum computers Kyriienko et al. (2021); Paine et al. (2023a;b); Heim et al. (2021).

However, previous work on solving differential equation on quantum computers assumes that the expectation of quantum circuits can be calculated directly. This is only possible when simulating

quantum circuits. When using actual quantum computers, expectations must be taken via repeated sampling, a cost so far mostly ignored in using quantum neural networks for differential equation solving.

Physics-informed regularisation involves taking the derivatives of neural network outputs with respect to input features. In classical neural networks, access to higher order derivatives can be optimised by mixing various modes of automatic differentiation, e.g. by considering forward-mode over reverse-mode automatic differentiation, or via Taylor-mode automatic differentiation Griewank & Walther (2008).

In contrast, for quantum neural networks, calculating derivatives with respect to circuit inputs typically involves a cost $\mathcal{O}(N^d)$, where $N$ is the number of data encodings into a quantum circuit, and $d$ is the order of derivatives. This scaling is similar to repeated use of forward-mode automatic differentiation in classical neural networks: a strategy known to be suboptimal.

Attempts to alleviate the cost of automatic differentiation have been introduced into quantum circuits Abbas et al. (2023), Bowles et al. (2023). However, in contrast to automatic differentiation in silicon-based computers, introducing better scaling for automatic differentiation in quantum neural networks requires architectural restrictions.

In this work, we derive favourable formulations of physics-informed loss functions amenable to run on quantum neural networks on actual quantum hardware. We construct line integrals of physics-informed losses from domain boundaries on which one can define appropriate loss functions. This formulation obviates the need for higher-order automatic differentiation when solving initial-value problems for ODEs, and thus is especially interesting as a candidate to solve differential equations on near-term quantum hardware. In addition to reducing the order of automatic differentiation required for ODEs, our methods also prevent the need to separately balance boundary contributions and interior loss contributions as in regular physics-informed optimisation. Our methods also regularise on the global properties of neural networks via line integrals, which introduces an alternative optimisation scheme to the local nature of point-wise gradient optimisation represented by conventional PINN optimisations (see Figure 1).

While we present our methodologies with quantum neural networks in mind, they are equally applicable to arbitrary neural networks regardless of the underlying hardware. We demonstrate applications of line-integral propagated classical and spiking neural networks in the Appendix A.3.

## 2 METHODS

### 2.1 FORMULATION

A general differential equation of a dependent variable $u : \mathbb{R}^n \to \mathbb{R}^d$ in an open set (domain) $\Omega \subset \mathbb{R}^n$ with a boundary $\partial\Omega$ can be written as follows:

$$\mathcal{N}u(x) = 0 \quad \text{if} \quad x \in \Omega, \qquad \mathcal{B}u(x) = 0 \quad \text{if} \quad x \in \partial\Omega \tag{1}$$

where $\mathcal{N}$ and $\mathcal{B}$ are (potentially non-linear) differential operators. Note that since we leave $\mathcal{N}$ to be an arbitrary differential operator, equation 1 represents a rich class of differential equations including all partial differential equations (PDEs) and ordinary differential equations (ODEs).

### 2.2 PHYSICS-INFORMED NEURAL NETWORKS

The solution of equation 1 can be approximated by a neural network $f_\theta(x) : \mathbb{R}^n \to \mathbb{R}^d$, where $\theta$ represent trainable parameters.

We define the probability distributions $\mathbb{P}_\Omega$ and $\mathbb{P}_{\partial\Omega}$ over $\Omega$ and $\partial\Omega$. and the samples from these distributions are referred to as collocation points. The optimisation of PINNs constructs weights $\theta^*$ using the following losses and optimisation problem:

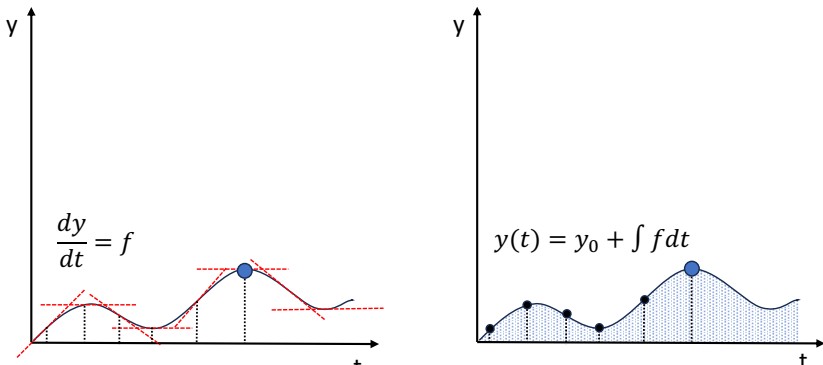

Figure 1: Left: Illustration of qPINNs. To train a variational quantum circuit (VQC) to predict a solution at a time T (blue dot), gradients need to be calculated, which is very costly with quantum neural networks. Right: Our proposed method requires no gradients (for initial value ODEs), but only requires the quantum circuit to be evaluated at points up until time T so that the area under the curve can be estimated. Since the right hand side relies on an averaging of points to the left of the blue dot to calculate the area, noise in each dot can be "averaged out" with noise from other dots, representing a further advantage in terms of optimising quantum neural networks with finite shots.

$$\begin{aligned}
\mathcal{L}_{\mathcal{B}}(\theta) &= \mathbb{E}_{x \sim \mathbb{P}_{\partial \Omega}}[\mathcal{B}(f_\theta(x))^2] \\
\mathcal{L}_{\mathcal{N}}(\theta) &= \mathbb{E}_{x \sim \mathbb{P}_{\Omega}}[\mathcal{N}(f_\theta(x))^2] \\
\theta^* &= \arg \min_\theta \mathcal{L}_{\mathcal{B}}(\theta) + \mathcal{L}_{\mathcal{N}}(\theta).
\end{aligned} \tag{2}$$

Note that hardly any restrictions have been placed on the architecture represented by $f_\theta(x)$ so far. Consequently, the optimisation problem in equation 2 applies equally to quantum neural networks as it does to classical neural networks. While in classical neural networks, the output of differential operators can be evaluated at collocation points via automatic differentiation Griewank & Walther (2008), the same can be achieved in quantum neural networks via the parameter-shift rule (PSR) Kyriienko & Elfving (2021); Izmaylov et al. (2021); Wierichs et al. (2022); Banchi & Crooks (2021); Schuld et al. (2019); Mitarai et al. (2018), which might be seen as a quantum version of forward-mode automatic differentiation.

For (digital) quantum neural networks, data and network weights are typically encoded via a rotation on a given qubit. Derivatives with respect to these encodings typically involve two additional circuit constructions per encoding. For $N$ data encodings, and $d$-th order derivatives, the number of circuits needed to evaluated the $d$-th order derivative thus scales as $\mathcal{O}(N^d)$, incurring significant number of circuit evaluations.

## 2.3 GLOBAL PHYSICS-INFORMED LOSSES

In section 2.3.1, we outline a method to derive global physics-informed losses in a general setting, with a particular emphasis on how to impose Dirichlet and Neumann boundary conditions. In section 2.3.2 we specialise these results specifically to ordinary differential equations, which represent a particularly compelling setting for quantum neural networks. While this work primarily considers quantum neural networks, the methodology sections are architecture-agnostic and thus apply equally to classical neural networks as they do to quantum neural networks.

### 2.3.1 PROPAGATED GLOBAL PHYSICS-INFORMED LOSSES

This section introduces a methodology to construct line integrals which propagate boundary conditions from the boundary, $\partial \Omega$ into the domain $\Omega$. Parameterise a solution to a differential equation by a neural network $f_\theta$, with trainable weights $\theta$. For an invertible differential operator $\mathcal{N}_B$, where

the suffix denotes its significance in propagating a boundary condition, equation 1 gives:

$$u = \mathcal{N}_B^{-1} \left[ (\mathcal{N}_B - \mathcal{N}) f_\theta \right], \tag{3}$$

which follows from noting $(\mathcal{N} - \mathcal{N}_B + \mathcal{N}_B) u = 0$ from equation 1, substituting $f_\theta$ for $u$ and then gathering terms.

Introduce a probability distribution, $\mathbb{P}_L$, of straight lines starting from points on $\partial\Omega$ going into $\Omega$, and a probability distribution $\mathbb{P}_l$ of points along a given line $l$. This allows for the following optimisation problem to be defined,

$$\theta^* = \arg\min_\theta \mathbb{E}_{l \sim \mathbb{P}_L} \left[ \mathbb{E}_{x \sim \mathbb{P}_l} \left[ \left( f_\theta(x) - \mathcal{N}_B^{-1} \left[ (\mathcal{N}_B - \mathcal{N}) f_\theta \right](x) \right)^2 \right] \right], \tag{4}$$

where $\theta^*$ is a parameterisation of $f_\theta$ which solves the differential equation.

Now it remains to define operators $\mathcal{N}_B$ which propagate boundary conditions from $\partial\Omega$ to $\Omega$.

**Dirichlet Boundary Conditions**  Choose a point, $x_0 \in \partial\Omega$ with a boundary condition $u(x_0) = u_0$. Parameterise a line $x_s = x_0 + sv$, where $s \in [0, S]$, $v$ is direction into $\Omega$ from $\partial\Omega$ and $x_s \in \Omega$ for all $0 < s < S$. Then parameterising the solution $u$ with a neural network $f_\theta$ and a choice of $\mathcal{N}_B = \frac{\partial}{\partial x_s}$ yields that:

$$u(x) \approx \mathcal{N}_B^{-1} \left[ (\mathcal{N}_B - \mathcal{N}) f_\theta \right](x) = u_0 + \int_{x_0}^{x} \left[ \mathcal{N}_B - \mathcal{N} \right] f_\theta(x') dx', \tag{5}$$

where $x$ is a point on the line and the integral represents a line integral along $x_s$ from $x_0$ to $x$. Now $u$ in the integrand can be parameterised with a neural network $f_\theta$ which then defines the Volterra integral equations to be used in the optimisation problem in equation 4.

**Neumann Boundary Conditions**  As in the previous section, consider the same point and parameterised line. However, instead of a Dirichlet boundary condition, we take $\nabla_v u|_{x_0} = u_0'$, where $\nabla_v$ denotes a directional derivative towards $v$. Then parameterising the solution $u$ with a neural network $f_\theta$ and a choice of $\mathcal{N}_B = \frac{\partial^2}{\partial x_s^2}$ gives:

$$u(x) \approx \mathcal{N}_B^{-1} \left[ (\mathcal{N}_B - \mathcal{N}) f_\theta \right](x) = u_0 + su_0' + \int_{x_0}^{x} \int_{x_0}^{x'} \left[ \mathcal{N}_B - \mathcal{N} \right] f_\theta(x'') dx'' dx', \tag{6}$$

where, as previously, $x$ is a point on the line and the integral represents a line integral along $x_s$ from $x_0$ to $x$. However, in this scenario, since our boundary is a Neumann condition, $f_\theta(0)$ should be substituted in for $u_0$ in equation 6 to avoid biasing the resulting loss function towards any specific Dirichlet condition. The resulting expression then defines the line integrals to be optimised upon in equation 4.

**Remarks**  The formulation of boundary propagated differential equation losses provides several benefits: 1. The boundary condition and interior terms—which are summed and weighted as separate terms in traditional PINN formulations—are combined into a single term. 2. Boundary conditions are propagated into $\Omega$ from $\partial\Omega$ via a loss derived directly from the underlying differential equation. This contrasts to PINNs which optimise points independently of one another. 3. Dependent on the specific forms of $\mathcal{N}_B$ and $\mathcal{N}$, we can achieve a reduction in the order of automatic differentiation in the optimisation of equation 4 compared to equation 2. In the case of initial-value problems in ODEs, higher-order automatic differentiation is not required at all. See section 2.3.2 for more details. 4. If the evaluation of $f_\theta(x_i)$ is stochastic, as in the measurement of a quantum neural network, then a (quasi) Monte-Carlo estimation of the integral averages noisy evaluations across the integral domain.

The integral in equation 5 must be evaluated numerically. This can be done by sampling points $x_i$ along the line $x_s$, computing $\left[ \mathcal{N}_B - \mathcal{N} \right] f_\theta(x_i)$ via automatic or numerical differentiation, and then numerically approximating the integral via Monte-Carlo integration, trapezoidal approximations, numerical quadratures, or quantum-enhanced Monte-Carlo integration Akhalwaya et al. (2023).

Given $L$ different lines emanating from $\partial\Omega$, each evaluated at $N$ different points, the computational complexity of the evaluation of the line integrals is $\mathcal{O}(NL)$. This is the same order of computational complexity as the evaluation of a forward-pass of a neural network at $NL$ different points. Finally, the integral in equation 6 represents a double line integral, as opposed to a surface integral. This can be calculated by repeating an integral estimation twice. Thus, the evaluation of Neumann boundary integrals of the form in equation 6, even though nested, remains of the order $\mathcal{O}(NL)$. The formulations of loss functions to impose Dirichlet and Neumann boundaries are not unique. For example, equation 6 can be used to regularise to Dirichlet boundary conditions by including Dirichlet boundaries for $u_0$ and substituting $f'_\theta$ for $u'_0$.

### 2.3.2 Shot-Efficient ODE Solving via Global Losses

While the methodologies in section 2.3.1 apply to a broad class of differential equations, we find a particular instance of that methodology to be of particular relevance to ODEs. Consider an ODE defined by:

$$\mathcal{N} = \frac{du}{dt} - g(u, t) = 0, \quad u(0) = u_0 \quad 0 < t < T, \tag{7}$$

with $u : \mathbb{R} \to \mathbb{R}^d$, $d \in \mathbb{N}$. Note that this formulation encompasses non-linear initial-value problems of arbitrarily high order since an $n$-th order initial-value problem can be rewritten as a set of $n$-coupled first-order initial-value problems. For example, $\frac{d^2 y}{dt^2} = g(y, t)$ can be written as the paired ODEs $\frac{dy}{dt} = z$ and $\frac{dz}{dt} = g(y, t)$.

Since our boundary condition in this case comprises of a single line, the $\mathbb{P}_L$ in equation 4 comprises a single line. We denote a probability distribution of time between $0$ and $T$ by $\mathbb{P}_t$. Parameterising a solution to equation 7 with a neural network $f_\theta$ and combining equation 5, equation 4 then yields the following optimisation problem to solve for the differential equation:

$$\theta^* = \arg\min_\theta \mathbb{E}_{t \sim \mathbb{P}_t} \left[ \left( f_\theta(t) - \left( u_0 + \int_0^t g(f_\theta(t), t) dt' \right) \right)^2 \right]. \tag{8}$$

Note that there are no derivatives of the $f_\theta$ with respect to its input, in contrast to the optimisation problem used in traditional PINN formulations outlined in equation 2. The optimisation problem in equation 8 can result directly from equation 7 by manually integrating with respect to time on both sides. However, the formulation provided in section 2.3.1 applies to a much larger class of differential equations.

The removal of higher-order neural network derivatives from the optimisation problem in equation 8 compared to regular PINNs is of especially large significance in solving differential equations with quantum circuits. Combining the lack of requirements of higher-order derivatives with gradient-free optimisation allows for a near arbitrary choice of hardware to solve ordinary differential equations.

Since finding the $d$th order derivative of a quantum neural network with $N$ feature-encodings requires $\mathcal{O}(N^d)$ evaluations, our method thus provides an $\mathcal{O}(N)$ reduction in the number of quantum neural network shots per training step.

## 3 Applications

This section presents the homogeneous, non-homogeneous and non-linear ordinary differential equations with applications in population modelling. We emphasise experimental setups achievable on near-term quantum hardware. While the methods in Section 2.3.1 apply to multiple types of differential equations and can be used to remove higher-order automatic differentiation requirements for some PDEs, this section demonstrates the applicability of method to ODEs.

Our methodologies also apply to almost arbitrary neural network architectures. Since our work is primarily concerned with shot-efficient solving of differential equations with quantum neural networks, we present results with multilayer perceptrons and spiking neural networks in the Appendix A.3.

We consider models of population extinction, population growth in a seasonal environment, and population evolution in an environment with limited resources, given respectively by the equations:

$$\frac{dy}{dt} = -\frac{3}{2}y, \quad \frac{dy}{dt} = 3\cos(3t)y, \quad \frac{dy}{dt} = y(1-y) \tag{9}$$

starting at time $t = 0$ with maximum times of $t = 2.0$, $t = 1.0$, $t = 5.0$, and with initial conditions $y = 1.5$, $y = 1.0$ and $y = 0.1$, respectively.

Investigations into the optimisation of quantum neural networks in finite shot training is still in its nascent stages. Since automatic differentiation over trainable parameters in quantum neural networks carries a computational overhead scaling linearly with the number of parameters, as opposed the constant scaling of backpropagation with classical neural networks, there is a stronger incentive to use gradient-free methods to train quantum neural networks. Thus, we compare gradient-based methods (with several learning rates) and gradient-free methods.

**Experimental Details** Experimental setups were chosen to ensure a degree of achievability on current quantum hardware. We consider a three-qubit circuit of the form $\langle \emptyset | U_t^\dagger U_\theta^\dagger H U_\theta U_t | \emptyset \rangle$, where $U_t$ encodes data via a Chebyshev-tower type feature map Kyriienko et al. (2021), $U_\theta$ represents a parameterised variational unitary comprising of single-qubit rotations followed by a triangular ring of entangling CNOT gates as in a hardware-efficient ansatz Kyriienko et al. (2021) Kandala et al. (2017). Input features were scaled linearly $[-0.95, 0.95]$ prior to the Chebyshev-feature encoding ensuring the arccos operation in the Chebyshev feature encoding is well-defined. Variational parameters, $\theta$, were initialised to be uniformly distributed in the range $[0, 2\pi]$. The Hamiltonian, $H$, is single-qubit magnetization on the zeroth qubit. The structure of the circuit used is shown in Figure 2.

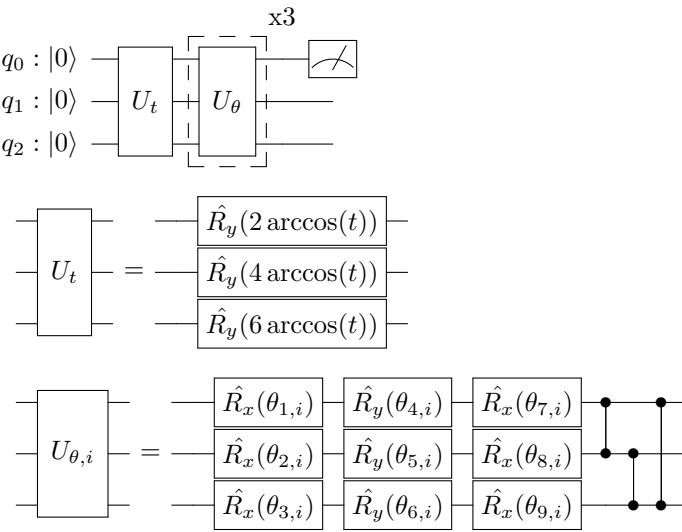

Figure 2: The quantum circuit diagram used for the shots experiments: (top) diagram shows the circuit diagram with three qubits q0, q1, q2, initialised with the zero basis state, a data encoding unitary $U_t$ that applies a Chebyshev-tower type feature map, three paramaterised variational unitaries $U_{\theta,i}$, that comprise the hardware-efficient ansatz (HEA), and measurement; (mid) data encoding unitary, and (bottom) the HEA in greater detail.

As a gradient-based optimiser, we use Adam Kingma & Ba (2014) with learning rates specified in the results section. For gradient-free optimisation, we use SPSA Spall (1992) (with hyperparameters standard deviation c=0.15, and a=2.0 needed for the calculation of the magnitude of the gradient) and One-Plus-One optimiser Droste et al. (2002) with budget=100, otherwise all remaining hyperparameters as set by Nevergrad Rapin & Teytaud (2018). We make use of PennyLane Bergholm et al. (2018), JAX Bradbury et al. (2018), optax Babuschkin et al. (2020) and Nevergrad in all our quantum experiments, and PyTorch Paszke et al. (2019) and snnTorch Eshraghian et al. (2021) for

Table 1: Number of shots of a quantum circuit required per training epoch for our proposed propagated methodology compared to existing qPINN methods. Note that the approximate factor of 6 appearing between the two methods corresponds to requiring two extra circuit evaluations required for each of three data-encodings in our quantum circuits. For $N$ data-encodings, our proposed method gives an $\mathcal{O}(N)$ factor reduction in the number of shots required per training epoch.

| Method | Gradient-based | Gradient-free |
|---|---|---|
| Propagated | 56,320 | 1,024 |
| qPINN | 394,240 | 7,168 |

our multilayer perceptron and spiking neural network simulations shown in Appendix A.3. The number of collocation points and the number of repeated measurements for given circuit parameters were both set to 32 for the finite shot experiments, with collocation points arranged in a uniform grid. Numerical integration for propagated losses was done via a quasi Monte-Carlo method across collocation points. Training is done over 10k epochs per method. While finite shots are used during training, solution quality was calculated on the basis of expectations taken on a uniform grid of 128 points over the temporal domain. All results are over 20 different random seeds. We compare physics-informed solutions for each system as given in equation 2, which we denote as *qPINN* to the propagated optimisation regime outlined in equation 8, which we denote as *propagated*.

Our experimental setup dictates a number of shots per training epoch dependent on the optimisation procedure used, which is summarised in table 3.

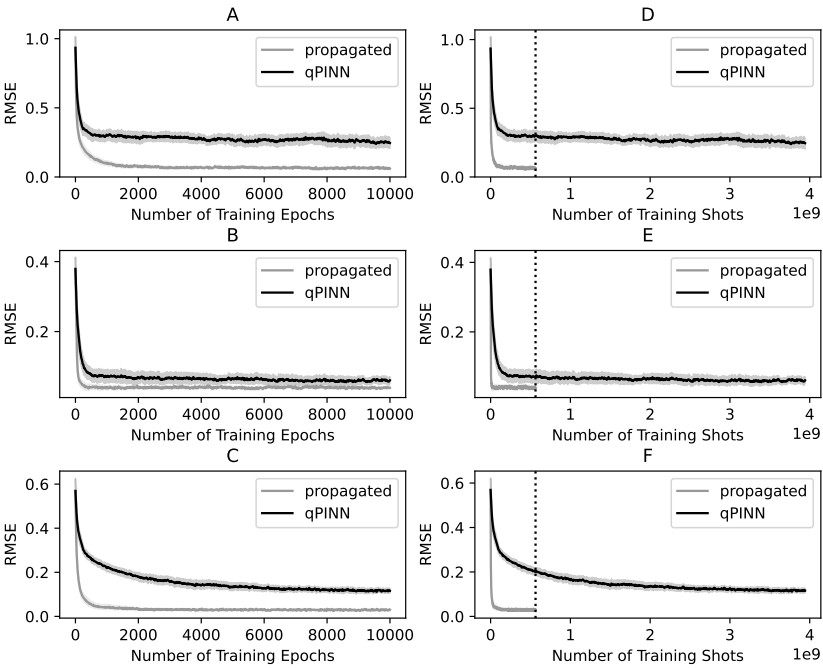

Figure 3: Mean qPINN (black) vs Global losses (gray) with shaded regions representing 95% confidence intervals of the mean calculated via bootstrapping over neural networks trained over 20 random seeds. We demonstrate lower root mean squared error (RMSE) in terms of both the number of training epochs (A-C) and training shots (D-F). Both models were trained using the Adam optimiser with learning rate 0.01. A|D) present the results for the seasonal growth of a population, depending on its environment, B|E) show the logistic growth model, and C|F) illustrate the population extinction model.

### 3.1 PROPAGATED LOSSES PROVIDE PROMISING MEANS OF SOLVING DIFFERENTIAL EQUATIONS ON NEAR-TERM QUANTUM HARDWARE

Across a full range of gradient-based optimiser settings and gradient-free optimiser settings, we find both faster convergence in terms of shot number and number of epochs (see Figure 3) and also lower end RMSEs. While Figure 3 shows a specific optimiser setting, its behaviour is representative of a broader trend across various learning rates and gradient or gradient-free optimisers. See the Appendix for further experiments A.2.

### 3.2 INFINITE SHOT QUANTUM NEURAL NETWORK ALGORITHM PERFORMANCE IS NOT NECESSARILY INDICATIVE OF FINITE-SHOT PERFORMANCE

Given the prevalence of infinite shot experiments, it is interesting to consider whether the performance of various algorithms in the limit of infinite shots translates to their performance with finite shots.

We observe that in some cases, algorithm performance in the infinite shot limit is not indicative of algorithm performance in the finite shot limit.

In Figure4, we demonstrate a scenario where a quantum PINN in the infinite shot limit demonstrates much lower RMSEs compared to propagated losses in the infinite shot limit, but much higher RMSEs in the finite shot limit.

The scientific relevance of this result is two-fold: 1. To illustrate that infinite shot behaviour of quantum neural networks is not necessarily illustrative of finite shot behaviour. 2. The favourable performance of our propagated loss formulation in the finite shot regime.

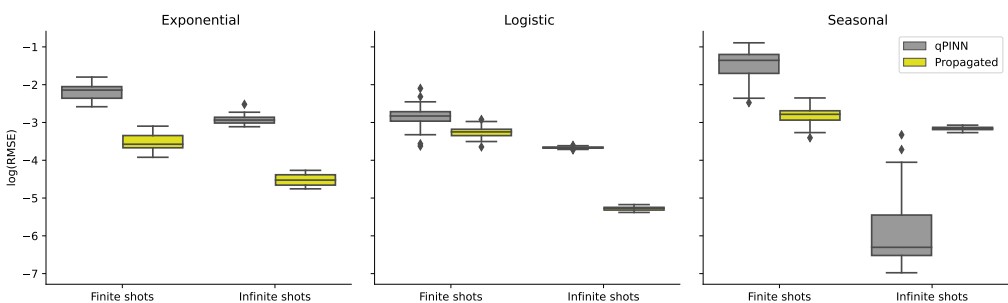

Figure 4: A comparison in the final values of the log root mean squared error (RMSE) achieved after 10000 training epochs between the finite (left) and infinite shots (right) cases for the propagated (yellow) and qPINN (grey) methods using the Adam optimiser with a 0.01 learning rate. The boxplots show the loss distribution for the 20 random seed experiments. Left to right subplots show the population extinction model, the logistic growth model, and the seasonal growth of a population, depending on its environment. Note in particular that for the seasonal model the qPINN shows preferable RMSE to propagated loss functions in the infinite shot limit, but worse RMSE in the finite shot limit.

## 4 CONCLUSIONS

This work demonstrates a means of solving ordinary differential equations with quantum neural networks in a manner amenable to near-term quantum hardware. The proposed methods performed favourably compared to benchmark solutions in terms of both shot requirements and RMSEs .

Experimentally, we find that the relative performance of quantum neural networks in the finite and infinite shot limits can be unpredictable, emphasising the need to consider finite-shots in quantum neural network when designing a quantum algorithm.

While we have demonstrated promise of our propagated loss formulation for ODE solutions, there remain many future directions of research. One intriguing direction involves exploring various collocation strategies for the propagated loss functions. Strategies for better sampling of collocation points are an active area of research in the field of PINNs Nabian et al. (2021); Guo et al. (2022) with some evidence that there is a relationship between the sampling strategies and the inability of the PINNs to converge to the correct solution (also called failure modes) Daw et al. (2023). Given that propagated physics-informed loss functions rely on numerical integration, it might be beneficial to draw inspiration from Monte-Carlo integration techniques, such as importance sampling, to enhance the presented techniques. We hope that our work demonstrates the importance of considering the implications of finite shots in quantum neural network algorithm design and motivates further research on this front.

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

# A  APPENDIX

## A.1  EXTENDED PHYSICS-INFORMED NEURAL NETWORKS METHODS

The proposed method of propagated global loss was compared to a PINN, constrained PINN, Augmented Lagrangian PINN on three different neural networks types—multilayer perceptron (MLP), quantum neural network (QNN), and spiking neural network (SNN) presented in section A.3. This section introduces the augmented PINN methods and the SNN architecture for scientific machine learning.

### A.1.1  THE AUGMENTED LAGRANGIAN METHODS

First, we introduce the Lagrangian method. Given some loss function $L(\theta)$ subject to constraints $C_i(\theta) = 0$, we might define some loss function like $L_{total} = L + \sum_i C_i^2$ to include the constraints. However, as we allude to above, these constraints might not be adhered to especially strongly.

To obtain stronger constraints, we might use Lagrange multipliers. We introduce a Lagrange multiplier $\lambda_i$ for each $C_i$ and solve the following optimisation problem:

$$\max_\lambda \min_\theta \left[ L + \sum_i \lambda_i C_i \right]. \tag{10}$$

In practice this means a different optimisation loop, taking alternating optimisation steps for $\lambda$ and $\theta$ at each overall training step.

Now, we can present the augmented Lagrangian method. Incorporating a straightforward implementation of Lagrange multipliers as above is only valid in cases where $L$ is locally convex, which unfortunately is not the case with NN and QNNs. When the problem is not locally convex, parameters $\lambda$ can start to diverge, leading to unstable training. Reintroducing the variational terms now re-stabilises training. See Bertsekas (1976) for a load more detail.

$$\max_\lambda \min_\theta \left[ L + \sum_i \lambda_i C_i + \beta_i C_i^2 \right] \tag{11}$$

where $\beta$ is a hyper parameter to be set, which take $\beta = 1$ for all experiments.

We follow Son et al. (2022), in assigning a Lagrange multilier $lambda_i$ for each collocation point. We consider two variants: one variant constrains only the boundary conditions via the augmented Lagrangian method (denoted *augmented-lagrangian* in the results), and another which constrains the interior differential equation loss via augmented Lagrangian method (denoted *augmented-lagrangian interior* in the results).

### A.1.2  SPIKING NEURAL NETWORKS (SNN)

Spiking neural networks (SNN) are a type of neural networks that resembles the biological interactions between neurons in the central nervous system Eshraghian et al. (2021). The transmission of spikes in SNN is highly time-dependent, with the spikes being binary signals compared to the continuous values outputs of NNs. As a general rule, the neuromorphic algorithms and hardware have a high energy efficiency that is beneficial for resource-constrained applications.

There is a myriad of proposed neuron types varying in complexity due to their level of descriptiveness of natural neurons (it is important to note that such models usually have no exact solutions), but one of the most commonly used ones in practice are the leaky integrate-and-fire (LIF). The LIF neuron can be described as a simplified biological neuron model that resembles a resistor (R) and capacitor (C) circuit as given in equation 12, where $\tau$, U, I, R are the membrane time constant, membrane potential, input current, and resistance. A spike (signal) is transmitted to other neurons is output only when the membrane potential U reaches a threshold value.

$$\tau \frac{dU}{dt} = -U + IR \tag{12}$$

The spiking neural networks have potential to be applied to the scientific machine learning domain to solve relevant problems in the form of differential equations in an energy-efficient manner Zhang et al. (2023).

## A.2 ADDITIONAL SHOTS EXPERIMENTS

These are additional visualisations of the finite and infinite shots experiments described in section 3. In the finite shots case, as stated, we used 32 shots, while in the infinite shots case the number of shots was set to None in PennyLane, which gives the exact expectation values as described in the documentation.

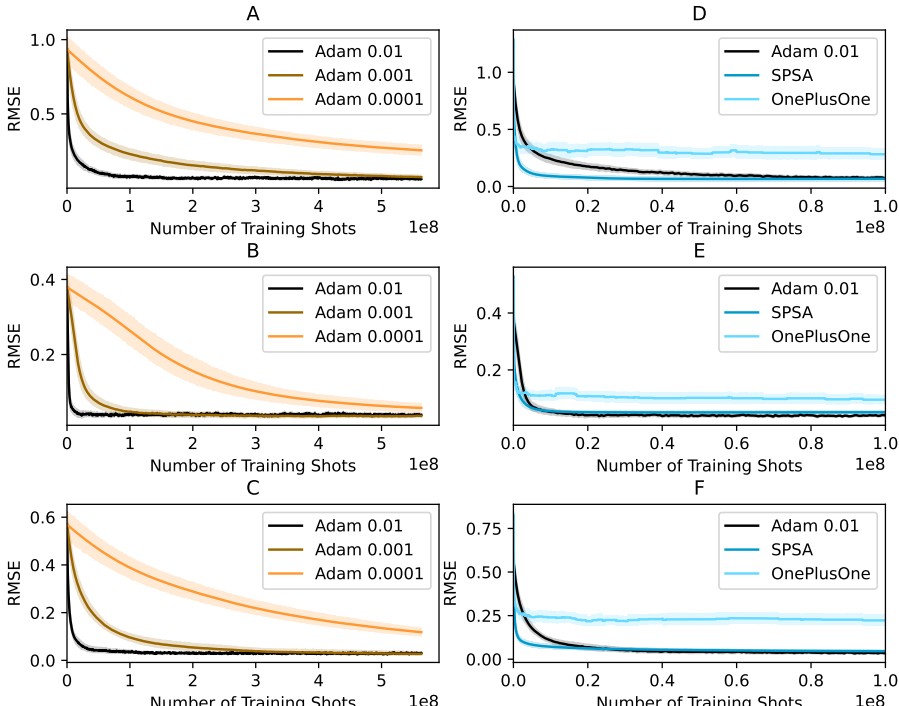

Figure 5: A comparison in the root mean squared error (RMSE) in terms of the number of training shots between the Adam optimiser with three different learning rates and the SPSA and One Plus One gradient-free algorithms in the finite shots case. The shaded regions representing 95% confidence intervals of the mean calculated via bootstrapping over neural networks trained over 20 random seeds. A|D) present the results for the seasonal growth of a population, depending on its environment, B|E) show the logistic growth model, and C|F) illustrate the population extinction model.

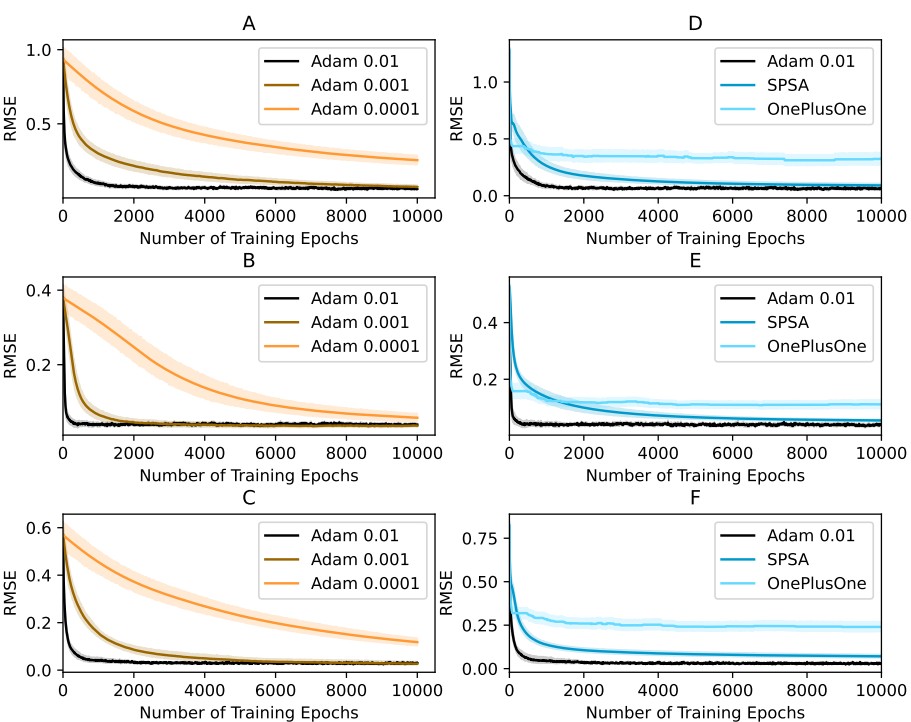

Figure 6: A comparison in the root mean squared error (RMSE) in terms of the number of training epochs between the Adam optimiser with three different learning rates and the SPSA and One Plus One gradient-free algorithms in the finite shots case. The shaded regions representing 95% confidence intervals of the mean calculated via bootstrapping over neural networks trained over 20 random seeds. A|D) present the results for the seasonal growth of a population, depending on its environment, B|E) show the logistic growth model, and C|F) illustrate the population extinction model

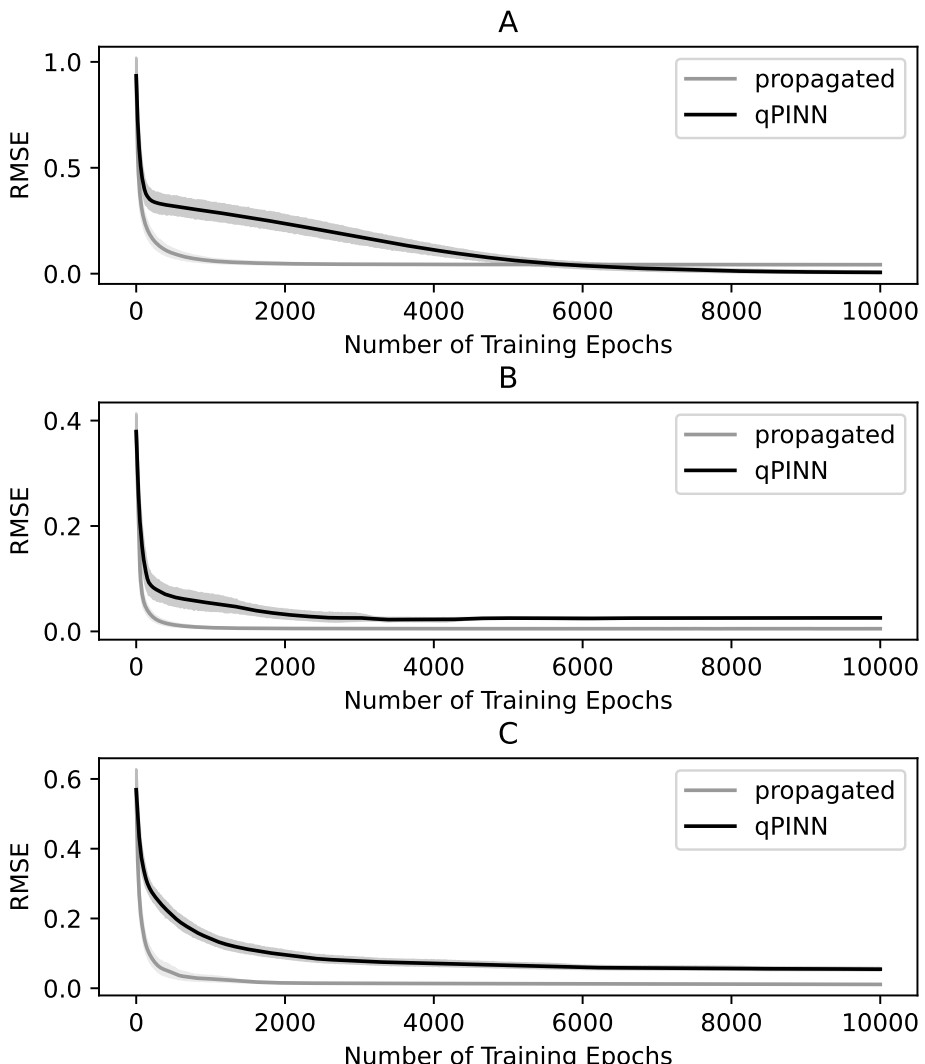

Figure 7: Mean qPINN (black) vs Global losses (gray) for a simple ODE with the shaded regions representing 95% confidence intervals of the mean calculated via bootstrapping over neural networks trained over 20 random seeds, demonstrating lower root mean squared error (RMSE) in terms of the number of training epochs in the infinite shots case. Both models were trained using the Adam optimiser with learning rate 0.01. A) presents the results for the seasonal growth of a population, depending on its environment, B) shows the logistic growth model, and C) illustrates the population extinction model.

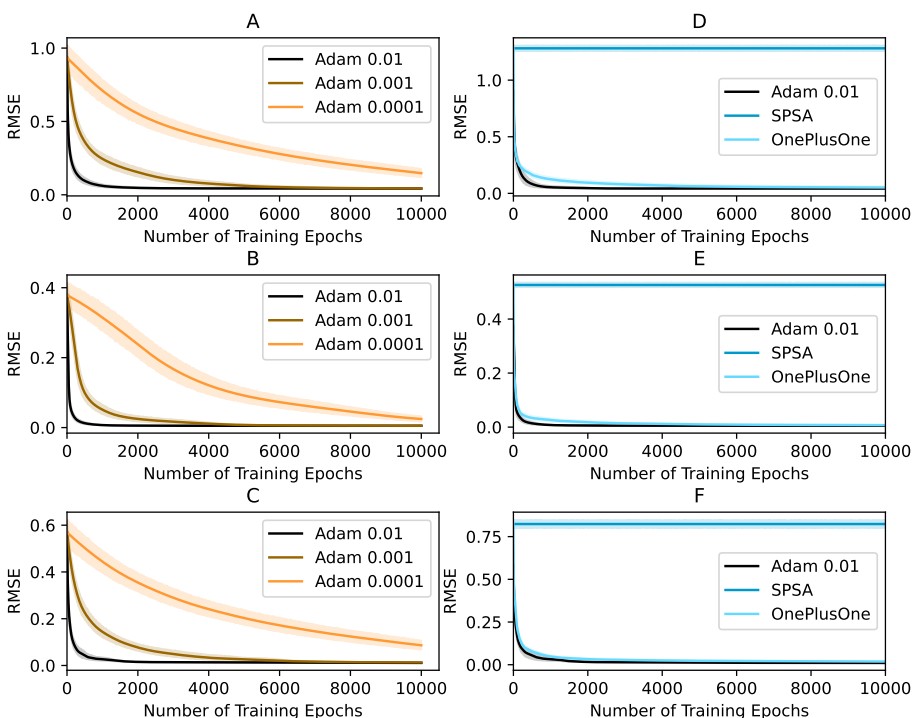

Figure 8: A comparison in the root mean squared error (RMSE) in terms of the number of training epochs between the Adam optimiser with three different learning rates and the SPSA and One Plus One gradient-free algorithms in the infinite shots case. The shaded regions representing 95% confidence intervals of the mean calculated via bootstrapping over neural networks trained over 20 random seeds. A|D) present the results for the seasonal growth of a population, depending on its environment, B|E) show the logistic growth model, and C|F) illustrate the population extinction model.

### A.3 ODE RESULTS

We conducted additional experiments solving four different physically and biologically important ODE examples, namely, the Brusselator, the Lorenz and Lotka-Volterra systems of equations, and the so called "parachute problem", or finding the final velocity of a falling body. Here, we were interested in comparing our new method labeled as propagated, to a vanilla PINN, a constrained PINN, and two Lagrangian-based methods that were given in greater detail in the previous section of the Appendix A.1.

We also considered three different types of neural networks: classical, a quantum, and spiking. The classical network has a feed-forward, multilayer perceptron (MLP) architecture with a single input (time data points), four hidden layers of 32 neurons each, and an output with a dimension matching the one of the ODE problem, using a hyperbolic tangent (tanh) as an activation function. The quantum network was already presented in the section 3. The SNN used 128 LIF neurons and has a single hidden layer and is based on the regression model presented in snnTorch Eshraghian et al. (2021).

The training of the MLP, QNN, and SNN was conducted using the Adam optimiser with 0.01, 0.001, and 0.0001 learning rates. The classical experiments were run for 32 000 epochs in the Lotka-Volterra and the Brusselator, or 64 000 for the parachute problem and the Lorenz system of equations. For the QNN experiments, we wanted to simulate realistically the current capabilities of the existing hardware and limited the training epochs to 1000, while for the SNN the training epochs were all set constant at 5000. All experiments are run using 20 random seeds to ensure reproducibility.

The results for the four main methods 1) propagated global loss method, 2) PINN, 3) constrained PINN, 4) Augmented Lagrangian PINN, labelled as *propagated*, *pinn*, *constrained*, *augmented lagrangian*, respectively (for more details see section A.1). The constrained PINN method uses exponential boundary pinning Lagaris et al. (1997), Lu et al. (2021).

#### A.3.1 THE BRUSSELATOR SYSTEM OF EQUATIONS

The Brusselator is as system of ODEs (see equation 14) describing an autocatalytic reaction for two chemical species x and y that depend on the supply of the substances a and b Prigogine (1980). It is one of the best studied chemical oscillator systems. The Brusselator is used as a typical example of a stiff equation when solved with the values for the substances at a=1, b=3. In our case, we solved the system in a stable regime with a=0.1, b=0.5, and initial values $x(0) = 1$ and $y(0) = 1$, for $t_{max} = 10$.

$$\frac{dx}{dt} = a + x^2y - bx - x \tag{13}$$
$$\frac{dy}{dt} = bx - x^2y$$

#### A.3.2 THE LORENZ SYSTEM OF EQUATIONS

The Lorenz system of equations comprises three ODEs and describes a chaotically behaving model of atmospheric convection Lorenz (1963), while being and nonlinear and deterministic. The equations are given in equation 15 with $\frac{dx}{dt}$ being the rate of change in convection, and $\frac{dy}{dt}$ and $\frac{dz}{dt}$ being the horizontal and vertical change in temperature, respectively, and $\sigma$, $\rho$, and $\beta$ are system parameters. We solved the Lorenz system of equations using the initial values $x(0) = 1$, $y(0) = 1$, $z(0) = 1$, and values for the parameters $\sigma = 10$, $\beta = 8 / 3$ and $\rho = 5$, for $t_{max} = 2$.

Table 2: Summary of the RMSE values per method and neural network type for the Brusselator. The RMSE mean and SD are calculated on the final values for the 20 random seeds. All models were trained using Adam with learning rate 0.001. The MLP, QNN, SNN were trained for 32 000, 1000, and 5000 epochs, respectively.

| Experiment | Method | Neural network | Mean | SD |
|---|---|---|---|---|
| brusselator | augmented-lagrangian | classical | 0.002976 | 0.003733 |
| brusselator | augmented-lagrangian | quantum | 0.902459 | 0.277743 |
| brusselator | augmented-lagrangian | spiking | 4.884190 | 3.952953 |
| brusselator | augmented-lagrangian-interior | classical | 0.203813 | 0.035461 |
| brusselator | augmented-lagrangian-interior | quantum | 0.962216 | 0.265557 |
| brusselator | augmented-lagrangian-interior | spiking | 3.781580 | 2.443145 |
| brusselator | constrained | classical | 0.004762 | 0.004572 |
| brusselator | constrained | quantum | 0.858994 | 0.274412 |
| brusselator | constrained | spiking | 2.609187 | 3.129017 |
| brusselator | pinn | classical | 0.002825 | 0.003758 |
| brusselator | pinn | quantum | 0.925948 | 0.278504 |
| brusselator | pinn | spiking | 3.576626 | 2.005476 |
| brusselator | propagated | classical | 0.007230 | 0.003229 |
| brusselator | propagated | quantum | 1.343192 | 0.232959 |
| brusselator | propagated | spiking | 1.591969 | 3.506904 |

Table 3: Summary of the RMSE values per method and neural network type for the Lorenz system of equations. The RMSE mean and SD are calculated on the final values for the 20 random seeds. All models were trained using Adam with learning rate 0.001. The MLP, QNN, SNN were trained for 64 000, 1000, and 5000 epochs, respectively.

| Experiment | Method | Neural network | Mean | SD |
|---|---|---|---|---|
| lorenz | augmented-lagrangian | classical | 0.002862 | 0.003027 |
| lorenz | augmented-lagrangian | quantum | 0.147204 | 0.101358 |
| lorenz | augmented-lagrangian | spiking | 3.488624 | 0.623335 |
| lorenz | augmented-lagrangian-interior | classical | 1.288524 | 0.000474 |
| lorenz | augmented-lagrangian-interior | quantum | 1.274591 | 0.201808 |
| lorenz | augmented-lagrangian-interior | spiking | 2.660279 | 1.124220 |
| lorenz | constrained | classical | 0.001751 | 0.001207 |
| lorenz | constrained | quantum | 0.044270 | 0.022340 |
| lorenz | constrained | spiking | 2.852119 | 0.853843 |
| lorenz | pinn | classical | 0.004316 | 0.003373 |
| lorenz | pinn | quantum | 0.311432 | 0.194034 |
| lorenz | pinn | spiking | 3.157912 | 0.798862 |
| lorenz | propagated | classical | 0.012058 | 0.001823 |
| lorenz | propagated | quantum | 0.138709 | 0.063387 |
| lorenz | propagated | spiking | 1.457266 | 1.201605 |

$$\frac{dx}{dt} = \sigma(y - x) \tag{14}$$
$$\frac{dy}{dt} = x(\rho - z) - y$$
$$\frac{dz}{dt} = xy - \beta z$$

Table 4: Summary of the RMSE values per method and neural network type for the Lotka-Volterra system of equations. The RMSE mean and SD are calculated on the final values for the 20 random seeds. All models were trained using Adam with learning rate 0.001. The MLP, QNN, SNN were trained for 32 000, 1000, and 5000 epochs, respectively.

| Experiment | Method | Neural network | Mean | SD |
|---|---|---|---|---|
| lotka-volterra | augmented-lagrangian | classical | 3.573486 | 0.141492 |
| lotka-volterra | augmented-lagrangian | quantum | 0.660898 | 0.200107 |
| lotka-volterra | augmented-lagrangian | spiking | 4.468683 | 6.338741 |
| lotka-volterra | augmented-lagrangian-interior | classical | 2.398411 | 0.005294 |
| lotka-volterra | augmented-lagrangian-interior | quantum | 2.256630 | 0.236550 |
| lotka-volterra | augmented-lagrangian-interior | spiking | 6.097998 | 6.549479 |
| lotka-volterra | constrained | classical | 0.000060 | 0.000133 |
| lotka-volterra | constrained | quantum | 0.727081 | 0.682835 |
| lotka-volterra | constrained | spiking | 3.612240 | 2.979569 |
| lotka-volterra | pinn | classical | 2.280876 | 1.718062 |
| lotka-volterra | pinn | quantum | 0.666538 | 0.192753 |
| lotka-volterra | pinn | spiking | 3.105012 | 1.002183 |
| lotka-volterra | propagated | classical | 0.574673 | 0.431178 |
| lotka-volterra | propagated | quantum | 1.147414 | 0.313880 |
| lotka-volterra | propagated | spiking | 1.411853 | 1.095471 |

### A.3.3 THE LOTKA-VOLTERRA SYSTEM OF EQUATIONS

We considered a Lotka-Volterra systems Lotka (1920), Volterra (1926) as a benchmark. This is a standard set of ODEs in Biology representing the time-evolution of a predator-prey system (see the equation below), that is widely used as a scientific machine learning benchmark, showing an oscillatory behavior Murray (2002). We solved the Lotka-Volterra equations with constant values a = 1.1, b = 0.4, c = 0.4, d = 0.1 and initial values x(0) = 5 and y(x) = 5, for $t_{max}$ = 10.

$$\frac{dx}{dt} = ax - bxy \tag{15}$$
$$\frac{dy}{dt} = dxy - cy$$

### A.3.4 THE PARACHUTE PROBLEM (TERMINAL VELOCITY OF A FALLING BODY)

Finding the terminal velocity of a falling body (frequently labelled as the parachute problem) serves as a classical physics textbook example of a mathematical model of Newtonian mechanics Meade & Struthers (1999) (see equation 16). We solved the parachute problem using the initial values v(0) = 0, and mass m = 70.0 kg, g = 9.81 m/$s^2$ and drag coefficient = 10.0, for $t_{max}$ = 10.

$$\frac{dv}{dt} = \frac{(mg - cv)}{m} \tag{16}$$

Table 5: Summary of the RMSE values per method and neural network type for the parachute problem. The RMSE mean and SD are calculated on the final values for the 20 random seeds. All models were trained using Adam with learning rate 0.001. The MLP, QNN, SNN were trained for 64 000, 1000, and 5000 epochs, respectively.

| Experiment | Method | Neural network | Mean | SD |
|---|---|---|---|---|
| parachute-problem | augmented-lagrangian | classical | 0.057748 | 0.061661 |
| parachute-problem | augmented-lagrangian | quantum | 0.111825 | 0.067211 |
| parachute-problem | augmented-lagrangian | spiking | 14.803368 | 5.130939 |
| parachute-problem | augmented-lagrangian-interior | classical | 0.035753 | 0.047889 |
| parachute-problem | augmented-lagrangian-interior | quantum | 0.057583 | 0.098441 |
| parachute-problem | augmented-lagrangian-interior | spiking | 26.809915 | 32.139628 |
| parachute-problem | constrained | classical | 0.022498 | 0.022315 |
| parachute-problem | constrained | quantum | 0.071080 | 0.140354 |
| parachute-problem | constrained | spiking | 18.933899 | 10.626157 |
| parachute-problem | pinn | classical | 0.070363 | 0.080651 |
| parachute-problem | pinn | quantum | 0.108740 | 0.063579 |
| parachute-problem | pinn | spiking | 23.560550 | 9.201258 |
| parachute-problem | propagated | classical | 0.934856 | 0.371374 |
| parachute-problem | propagated | quantum | 0.962259 | 0.370997 |
| parachute-problem | propagated | spiking | 8.915335 | 3.896813 |

