# OpenReview forum: "A Shot-Efficient Differential Equation Integrator using Quantum Neural Networks"
_ICLR.cc/2024/Conference — ICLR 2024 Conference Withdrawn Submission_

### Official Review · Reviewer_Nx77 · 2023-10-23

**Soundness:** 2 fair
**Presentation:** 2 fair
**Contribution:** 2 fair
**Rating:** 3
**Confidence:** 3

**Summary:**

The paper introduces a global physics-informed loss function that avoids the need for higher-order automatic differentiation, which is costly and challenging for QNNs. The paper shows that the proposed method can reduce the number of shots required to train QNNs for solving ODEs by an order of magnitude compared to existing q-PINN methods.

**Strengths:**

1. The authors propose a novel method for solving differential equations using quantum neural networks on near-term quantum hardware, which can reduce the number of shots required to train QNNs by an order of magnitude compared to existing methods.

2. It introduces a global physics-informed loss function that avoids the need for higher-order automatic differentiation, which is costly and challenging for QNNs.  The numerical experiments on simulated quantum circuits demonstrate the effectiveness and efficiency of the proposed method.

**Weaknesses:**

1. The paper lacks a clear motivation and contribution to the field of quantum machine learning. It does not explain why solving differential equations with quantum neural networks is important or novel, and how it compares to existing methods in classical or quantum computing.

2. It does not provide sufficient theoretical analysis or justification for the proposed method of global physics-informed losses. It does not show how the method is derived from general principles, what assumptions or limitations it has, and how it guarantees the convergence or accuracy of the solution.

3. The paper does not present any rigorous experimental results or benchmarks to demonstrate the effectiveness or efficiency of the proposed method. It only shows some qualitative plots of the solutions for three simple ODEs, without any quantitative metrics, error analysis, or comparison with other methods especially the classical learning-based methods for solving ODEs. It also does not report any details on the implementation, such as the number of qubits, circuit depth, optimization algorithm, hyperparameters, etc.

**Questions:**

Please see the weaknesses.

---

### Official Review · Reviewer_bf4n · 2023-10-29

**Soundness:** 2 fair
**Presentation:** 2 fair
**Contribution:** 2 fair
**Rating:** 3
**Confidence:** 4

**Summary:**

This work exploits an efficient physics-informed regularization by constructing a global formulation of physics-informed losses that are amenable to solving ODE on NISQ devices. In particular, the formulation incorporates boundary conditions and physics-informed losses into a single optimization term. The empirical simulation demonstrates the advantages of the proposed method against the PINNs.

**Strengths:**

1. The author leverages a means of ODE for quantum neural networks in a manner amenable to NISQ devices.

2. The proposed methods demonstrate better performance than the existing PINNs.

**Weaknesses:**

1. The formulation of the physics-informed regularization and the related ODE approach are not well presented. Many necessary and related backgrounds are missing in the article, which makes the paper hard to understand.

2. The proposed method claims there are no gradients, which are gradient-free, but the simulation includes both gradient-based and gradient-free for this method, causing much confusion about the proposed approach.

3. The cost analysis of the Dirichlet and Neumann Boundary conditions is not shown in the paper, so a trade-off between performance gains and computational costs.

4. The relationship between differential equations and the benefits of quantum neural networks is not clear at all.

5. The experimental setups, as shown in Figure 2, are taken for granted without an explicit explanation. For example, why there are two repeated R_{x} gates on the same quantum channel in composing U_{\theta, i}?

6. The use of RMSE as the loss function is not optimal, both MAE and MSE can ensure to attain better performance for regression problems.
Ref. Qi, J., Du, J., Siniscalchi, S.M., Ma, X. and Lee, C.H., 2020. On mean absolute error for deep neural network based vector-to-vector regression. IEEE Signal Processing Letters, 27, pp.1485-1489.

**Questions:**

1. Why did the authors choose the RMSE as the loss function rather than MAE and MSE?

2. What is the computational cost of the proposed Dirichlet and Neumann Boundary conditions to calculate the ODEs?

---

### Official Review · Reviewer_k8bX · 2023-10-30

**Soundness:** 3 good
**Presentation:** 2 fair
**Contribution:** 2 fair
**Rating:** 3
**Confidence:** 4

**Summary:**

This paper presents a new method for solving differential equations using quantum neural networks. This method is based on "a global formulation of physics-informed losses". It appears that, with this new formulation of empirical loss, this method can avoid higher-order auto-differentiation for initial value problems (ODE) and some PDEs. Also, the new method does not require a separate effort to handle the boundary value conditions by merging the boundary value conditions into the formulation of the empirical loss. It is also worth noting that this new formulation is not limited to quantum neural networks, but is also applicable to ordinary neural networks (empirical results are given in the appendices).

Beyond theoretical formulation, numerical simulation has been performed to solve three ODE models, namely, population extinction, population growth, and a seasonal environment. In each model, the problem is to solve a scalar function $y(t)$ governed by an ODE with initial conditions. The quantum circuit uses 3 qubits. The circuit design (i.e., feature map, hardware-efficient ansatz) almost completely follows Kyriienko et al. (2021). The function $y(t)$ is computed by measuring the first qubit with a Pauli operator (see Fig 2). The authors report the advantage of their method (*propagated*) compared to *qPINN*, especially when the number of shots is finite.

**Strengths:**

The main contribution of this work is a new formulation of physics-informed loss function via line integrals, as given in Equation (4). Under Dirichlet boundary conditions or Neumann boundary conditions, the loss function can be further simplified and evaluated by standard numerical or Monte Carlo integration techniques (e.g., Eq. (5) and (6)). It is argued that the computational complexity of the evaluation of the line integrals is $O(NL)$, where $L$ is the number of line elements, and $N$ is the number of quadrature points on each line. Using the new loss function with quantum neural networks, there is an $O(N)$ reduction in the number of shots per training step.

**Weaknesses:**

The presentation (both language & math) of this paper needs to be improved. Here, I only list a few questions regarding the mathematical expression & rigor:

1. In Equation (3), the authors introduce a symbol $\mathcal{N}_B$ without any prior definition. This symbol seems to be equivalent to $\mathcal{B}$ in Equation (1). However, the authors assume that $\mathcal{N}_B$ is invertible (in Section 2.3.1), which is a bit problematic. The inverse of differential operators such as $\frac{\partial}{\partial x}$ (in "Dirichlet Boundary Conditions") and $\frac{\partial^2}{\partial x^2}$ (in "Neumann Boundary Conditions") appear to be singular operators without a well-defined inverse.

2. In the quantum loss function $\langle \emptyset |U^\dagger_t U^\dagger_\theta H U_\theta U_t|\emptyset\rangle$, the authors use $\emptyset$ to denote the initial state, which is very confusing. Also, it would be much clearer if more explanations on how to use this ansatz to estimate the formulated loss function (Eq. (8)) were provided.

3. The authors use the term "finite and infinite shot limits" several times. I can understand the "infinite shot limit", which means the perfect expectation value of the measurement quantum observable. But I am not sure what is the "finite shot limit". Mathematically, a *limit* is defined as the value that a function (or sequence) approaches as the input (or index) approaches some value. When the number of shots is finite, I do not think the measurement result converges. Therefore, I do not understand what is a "finite limit" in the context of this paper.

**Questions:**

Besides the questions in the "Weakness" section, I want to ask why we have to use QNN to solve differential equations. The numerical examples are very simple scalar ODEs that do not require advanced techniques such as NNs to solve. The motivation for introducing physics-informed NN in numerical analysis is to deal with the "curse of dimensionality", as the traditional methods have bad scaling in terms of the problem dimension (especially for PDEs). However, I do not see why quantum could help this situation (and even do better than classical NN). As the author mentioned in the manuscript, QNN suffers from expensive (higher-order) gradient estimation and does not allow a perfect evaluation of the loss (which would require "an infinite amount of shots"). Classical NN does not have these issues. Also, the tables given in Appendix A.3 show that QNN does not outperform classical NN.

---

### Official Review · Reviewer_1rNZ · 2023-11-01

**Soundness:** 3 good
**Presentation:** 2 fair
**Contribution:** 2 fair
**Rating:** 5
**Confidence:** 3

**Summary:**

The paper proposes a learning-based PDE solver which obviates the need for high-order derivatives. This advantage is specifically important for quantum machine learning since evaluating high-order derivatives of parameterized quantum circuits or quantum neural networks requires high-order polynomial shots of quantum circuit execution. To achieve this, the authors propose an integral-based loss function and employ a Monte-Carlo approach to optimize the model. The method is evaluated on several differentiable equations with noiseless quantum simulators which showcase the advantage of savings in the shot number in training.

**Strengths:**

The finite-shot regime of quantum machine learning is an important yet less discussed problem in practical experiments. The paper demonstrates that we can design shot-efficient algorithms to leverage the need for evaluating multiple circuits. This fact is also supported by the experiment showcasing that the loss function decreases faster compared to the other method when using the same training shots.

**Weaknesses:**

The proposed method shifts the computation toward several hyper-parameters related to Monte Carlo integration and the number of lines to change boundary conditions to interior terms. However, the criteria for choosing the values of these parameters, which closely relate to the computation complexity, are not thoroughly discussed. These values are likely to be highly related to the performance of the algorithm, where a trade-off between efficiency and performance should be discussed.

The experiments are conducted on three toy ODEs (1-D problems), with a small variational ansatz. These examples are not convincing enough to support the major claims. It is likely that in high-dimensional problems with deeper ansatz, the proposed method requires a significantly larger L and more sampled points.

The presentation of the paper is also a bit messy. I.e., why do sections 3.1 and 3.2 lie inside applications? Why do the citations not have brackets over authors?

In section 2.3.2, the paper claims an O(N) reduction in the number of shots. Where does this come from? Does it mean that the number of shots is O(N^(d-1)) for the proposed method?

Minor points:
* Page 2 bottom: “$\partial \Omega. and the samples” unnecessary period.

**Questions:**

See the weaknesses.